# Using social marketing for the promotion of cognitive health: a scoping review protocol

Mathilde Barbier ,[1] Caroline Schulte,[2] Anna Kornadt,[1] Carine Federspiel,[3] Jean-Paul Steinmetz,[3] Claus Vögele[1]

¹Department of Behavioral and Cognitive Sciences, University of Luxembourg, Esch-sur-Alzette, Luxembourg
²Department of Computer Science, Therapeutic Sciences, University of Applied Sciences Trier Department of Computer Science, Trier, Germany
³Centre for Memory and Mobility, Zitha, Luxembourg city, Luxembourg

**Correspondence to**
Dr Mathilde Barbier;
mathilde.barbier@uni.lu

## ABSTRACT

**Introduction** The use of social marketing strategies to induce the promotion of cognitive health has received little attention in research. The objective of this scoping review is twofold: (i) to identify the social marketing strategies that have been used in recent years to initiate and maintain health-promoting behaviour; (ii) to advance research in this area to inform policy and practice on how to best make use of these strategies to promote cognitive health.

**Methods and analysis** We will use the five-stage methodological framework of Arksey and O'Malley. Articles in English published since 2010 will be searched in electronic databases (the Cochrane Library, DoPHER, the International Bibliography of the Social Sciences, PsycInfo, PubMed, ScienceDirect, Scopus). Quantitative and qualitative study designs as well as reviews will be considered. We will include those articles that report the design, implementation, outcomes and evaluation of programmes and interventions concerning social marketing and/or health promotion and/or promotion of cognitive health. Grey literature will not be searched. Two independent reviewers will assess in detail the abstracts and full text of selected citations against the inclusion criteria. A Preferred Reporting Items for Systematic Reviews and Meta-Analyses flowchart for Scoping Reviews will be used to illustrate the process of article selection. We will use a data extraction form, present the results through narrative synthesis and discuss them in relation to the scoping review research questions.

**Ethics and dissemination** Ethics approval is not required for conducting this scoping review. The results of the review will be the first step to advance a conceptual framework, which contributes to the development of interventions targeting the promotion of cognitive health. The results will be published in a peer-reviewed scientific journal. They will also be disseminated to key stakeholders in the field of the promotion of cognitive health.

## Strengths and limitations of this study

► Our method places emphasis on the feasibility of the search strategy to merge different concepts successfully.
► This will be the first scoping review describing existing electronic research and to determine the benefits of social marketing to affect knowledge and behavioural intentions in the promotion of cognitive health.
► The review will follow a rigorous methodology and map the evidence located in seven well-referenced databases.
► The time frame will be limited from 2010 onwards and to academic literature.
► We will not formally evaluate the evidence; however, this review will be the first step to conduct a full systematic review on the topic.

## INTRODUCTION

It is currently estimated that by 2050 the proportion of people aged 65+ years worldwide will have more than doubled.[1] Considering that age is a major risk for dementia,[2] the number of people affected by dementia is, at the same time, expected to increase to 131.5 million by 2050.[3] Previous reviews have concluded that pharmacological treatment to prevent or delay cognitive impairment[4] or to manage dementia,[5] is unpromising. There is recent evidence suggesting that combined therapy (eg, oral citicoline plus memantine) compared with one single treatment (eg, memantine) can slow disease progression in persons with Alzheimer's disease and mixed dementia.[6] These results are very promising from a medical and clinical point of view. From a prevention and thus public health perspective, however, knowledge on the effects of cognitive health promotion, especially among older adults, would be important but is relatively scarce.

The term 'cognitive health' is relatively new. People are less familiar with it and there is a need to cover the definition and scope of this term. Cognitive health has been defined as 'the ability to clearly think, learn and remember'.[7] The promotion of cognitive health can be considered as enabling people to 'increase control over their own [cognitive] health […] [through] a wide range of social and environmental interventions'.[8]

Potentially modifiable risk factors for cognitive impairment have been described in the literature.[9] They include alcohol consumption, depression, diabetes, hypertension, obesity, smoking and social isolation. One way of targeting these risk factors is through health promotion interventions.[8] There is growing consensus worldwide that maintaining and promoting cognitive health has become essential for healthy ageing.[10]

Recent recommendations[11] encourage the use of non-pharmacological interventions and prioritise the implementation of longitudinal studies when evaluating cognitive rehabilitation and/or cognitive stimulation therapy. They also call on community-based cognitive interventions provided by caregivers. This requires both training caregivers and making older people and their families aware of the importance to participate in these interventions.[12]

The ACTIVE study[13] was the first large-scale, multisite randomised controlled trial to compare the long-term effects of three cognitive training interventions for maintaining independence in older adults. The results suggested that cognitive training is a protective factor, thus reducing the prevalence of dementia. Subsequently, further initiatives were introduced for the promotion of cognitive health. Pilot studies have tested the implementation of programmes enabling healthy seniors to learn a new language,[14] or a dual-tasking tool enabling older adults (with or without mild cognitive impairment) to train their cognitive and physical functions at the same time.[15] A web-based tool was developed for educating older people on the promotion of cognitive health and helping them improve their health-related behaviors.[16] A 24-week multicomponent intervention programme was implemented in older adults with mild cognitive impairment to promote community activity using community resources.[17] The programme included physical, cognitive and social activity. The results demonstrated significant intervention effects on both cognitive and physical functions. These recent initiatives look promising for the promotion of cognitive health. Nevertheless, further research is needed, for example on the long-terms effects.[17 18] In addition, there remains a substantial gap in how to turn recommendations into practice for the promotion of individuals' cognitive health.[19]

Designing interventions with the aim of promoting health may be a complex challenge for a practitioner or a researcher.[20] Wold and Mittelmark[21] have argued in favour of expanding health promotion as a transdisciplinary research field. In this regard, social marketing constitutes one potential approach guiding the development of health promotion interventions.[22–24] Social marketing aims at (i) identifying marketing principles that can be applied to reach a prosocial goal and (ii) providing a useful theoretical framework for designing interventions and services that will contribute to the development of prosocial behaviours.[25] Thus, a social marketing strategy consists of a creation and administration of a marketing plan to effectively attract, motivate and retain an audience in performing a targeted prosocial behaviour. In the field of health, marketing has been defined as a person-centred approach by aligning health (promotion) services with specific characteristics and needs of the targeted populations.[26]

Social marketing offers strategies[27] that may be used to address wider determinants of health and to develop efficient public health interventions. Segmentation is one of those strategies allowing to identify healthcare consumer profiles.[28] Tailoring is another important strategy to adapt a service or an intervention to segments of the targeted audience. A recent study showed that tailored communication was more effective in increasing health screening behaviours in primary prevention than non-tailored communication.[29] The marketing mix, historically known as the 4Ps—Product, Price, Place and Promotion—can also be helpful. This strategy has been used for example as an approach in public health to categorise the existing nutrition policies, assess their perceived effectiveness and set guidelines for future policy actions.[30] With regard to health promotion, a recent systematic review has shown that the use of social marketing strategies was effective in increasing participation in, or the level of physical activity among older adults.[31]

The Behaviour Change Wheel[32] (BCW) is a model created from a review of frameworks for social marketing. Based on this model, it is possible to identify, for the first time, how elderly people perceive their capability, opportunity and motivation to adopt cognitive health promoting behaviours. This market analysis can be obtained from in-depth interviews. Then, a self-reported questionnaire, aligned with the findings from the interviews, will allow to link the capability, opportunity and motivation to the levers of action that should be addressed for each of these components. These levers are categorised as: (i) intervention functions, for example, education, incentivisation, environmental restructuring, modelling or enablement and (ii) policy categories, for example, social planning, service provision, regulation or legislation. Social marketing strategies may concern implementation intentions to increase capability, and ultimately, motivation and participation of older people in behaviour promoting cognitive health. They may also concern the use of public media to increase opportunity for the older people who lack knowledge to get familiar with this domain of their health, and to identify available resources in their local area that will help them maintaining their cognitive health. A last but not least strategy may concern the use of a social identity approach, for example, through positive social norms, to enhance the motivation of older people health-promoting behaviour with regard to their cognitive health. The BCW is one of the ways we can target and segment cognitive health promotion interventions according to the audience's needs. Social marketing is able to provide the healthcare sector and the government with realistic goals and action strategies in promoting cognitive health among older adults. Efficacy will be measured directly by monitoring

the number of people using the services (programmes and interventions) designed to promote cognitive health. A cumulative positive effect can also be observed. If older people use these services more and feel empowered to do so, they will be potentially more motivated and committed to continue using these services, and hence to promote the services among their acquaintances and family.

The potential of social marketing research has not yet been fully developed. We need more evidence to learn how social marketing strategies can be interconnected to further advance the promotion of health in general, and the promotion of cognitive health, in particular. We also need to know how this better understanding can further inform the development and implementation of effective population-based programmes and interventions. Scoping reviews are well-suited for providing a comprehensive overview on a particular topic.[33] One of its greatest strengths is to embrace sources of evidence as a whole. Scoping reviews are a multi-method[34] approach. They aim to provide an integrative synthesis, which is limited to a given topic.[35] They follow a rigorous and replicable scientific method.[34 36] Due to the heterogeneity in the sources of evidence included, one main limitation, however, is that scoping reviews do not formally assess quality.[36 37] Therefore, scoping reviews might be more useful to assess information prior to a full systematic review rather than to inform practice.[38]

A preliminary search of PubMed and DoPHER was conducted using the terms "cogniti*" AND "health*" AND "market*". We found neither scoping nor systematic reviews on this topic. In line with the rationale for conducting scoping reviews, we will map the evidence to explore whether health marketing strategies can inform the efficient implementation and use of services aiming at the promotion of cognitive health. A main goal is to identify in the literature the potential and the gaps concerning this topic. A related goal is to provide the scientific community and policymakers with future research directions. The scoping review will systematically review the literature in three concept groups: (i) social marketing, (ii) health promotion, (iii) cognitive health. The first concept group concerns the identification of social marketing strategies that have been shown to be effective in changing people's health behaviours or attitudes. In the second concept group we will explore the methods and approaches that have been used to successfully promote healthy lifestyle behaviours in various areas. The third concept group will focus on the methods and approaches that have been used to promote cognitive health.

## METHODS
### Patient and public involvement
Neither patients nor the public were involved in developing the present protocol.

The protocol has been developed using the five-stage methodological framework developed by Arksey and

O'Malley,[36] and updated by Levac *et al*[34] and the Joanna Briggs Institute.[39] The stages consist of: (i) identifying the research question(s), (ii) identifying relevant studies, (iii) study selection, (iv) charting the data and (v) collating, summarising and reporting the results.

### Stage I: identifying the research question(s)
The scoping review will address the following questions:
► Which social marketing strategies have already been designed and implemented in a public health setting?
► Which populations and contexts were targeted in health marketing programmes and interventions?
► Have health social marketing interventions resulted in positive outcomes in empowering people?
► What programmes and interventions (including their goals and characteristics) have been designed and/or implemented to promote cognitive health?
► What are the challenges for research and practice in merging social marketing strategies with the promotion of cognitive health?

### Stage II: identifying relevant studies
#### Search strategy
The study will be conducted from 1 November 2021 to 31 June 2022. As a preliminary step, all the publications in English prior to 2010 will be checked electronically in the Cochrane Library. Then, articles published in peer-reviewed scientific journals will be searched in the following web databases: the Cochrane Library, DoPHER, the International Bibliography of the Social Sciences, PsycInfo, PubMed, ScienceDirect, Scopus. Based on our preliminary searches, and due to the constraints in time and resources available, only publications in English and from 2010 to the present will be searched during this stage. Two reviewers (MB and CS) will perform the literature search independently.

Table 1 gives a summary of the search terms to be used in each of the three concept groups related to social marketing, health promotion and cognitive health. Combinations of different terms will be implemented by using the Boolean operator OR.

The text words contained in the titles and abstracts of relevant articles, and the index terms used to describe the articles will be used to develop a full search. Finally, the reference lists of articles included in the review will be screened for additional papers.

### Stage III: selection of studies
Two reviewers (MB and CS) will screen the titles and abstracts independently. All identified records will be collated and uploaded into Mendeley, and duplicates will be removed. Once screening is complete, inter-rater reliability will be assessed using Cohen's Kappa statistic and results will be discussed between the two reviewers. Any disagreement that cannot be resolved will be submitted to a third person of the research team for resolution.

The full text of selected citations will then be assessed in detail against the inclusion criteria by the two reviewers

**Table 1** Search concepts and terms used

| Concept 1: social marketing | Concept 2: health promotion | Concept 3: cognitive health |
|---|---|---|
| "marketing" OR<br>"market* approach*" OR<br>"marketing mix" OR<br>"market* segment*" OR<br>"market* strateg*" OR<br>"social marketing" OR<br><br>'"tailoring" | "health"<br>"health* behaviour*" OR<br>"health* behavior*"<br>"healthy lifestyle" OR<br>"health promot*" OR<br>"health-promoting" OR<br>"prevent*" OR<br>"public health" | "brain health" OR<br>"cognitive" OR<br>"cognitive abilit*" OR<br>"cognitive funct*" OR<br>"cognitive health" OR<br>"cognitive skill*" OR<br>"healthy ageing" OR<br>"healthy aging" OR<br>"healthy brain" |

independently. Inter-rater reliability will also be assessed at this stage using Cohen's Kappa, and any doubt, question or disagreement will be resolved through discussion or by calling in a third party, if necessary. Reasons for exclusion of full-text articles that do not meet the inclusion criteria will be recorded and reported in the scoping review.

The inclusion and exclusion criteria are presented in table 2 using the population, concept and context eligibility criteria for scoping reviews.

We will consider studies that include the design of and the evidence concerning (i) social marketing interventions targeting the promotion of health and (ii) interventions targeting the promotion of cognitive health. Studies performed in any country will be of interest, and studies conducted in any setting will be considered. We will include both primary and intervention research. We will examine different types of evidence reviews (eg, 'state of the art' reviews, conceptual reviews, realistic reviews, critical reviews, narrative reviews), and we will look at

qualitative, quantitative and multi-methods studies. Protocols published in peer-reviewed scientific journals will also be included. Any change in inclusion and exclusion criteria will be reported in the full scoping review.

The results of the search will be reported in full in the final scoping review and presented in a Preferred Reporting Items for Systematic Reviews and Meta-analyses for Scoping Reviews flow diagram.[40]

### Stage IV: extracting and charting the results

Data will be extracted from papers included in the scoping review using a data extraction form adapted from the Cochrane Collaboration.[41] A copy of this form is available as an online supplemental file. The data extracted will include specific details about the population and settings, methods, participants, procedure, results and applicability relevant to the review questions. The draft data extraction tool will be modified and revised as necessary during the process of extracting data from each paper included. Changes will be detailed in the full scoping review. Data

**Table 2** Inclusion and exclusion criteria

| | Inclusion criteria | Exclusion criteria |
|---|---|---|
| Population | ► Adults from any age group directly targeted by programmes and interventions aiming at the promotion of (cognitive) health.<br>► Family and community of the target audience as well as physicians, caregivers and health practitioners who are involved in designing and implementing interventions. | ► People under the age of 18 years.<br>► Animals. |
| Concept | ► Articles reporting the design, implementation, outcomes and/or evaluation of programmes and interventions aiming at the promotion of (cognitive) health, and with or without social marketing alignments.<br>► Qualitative and quantitative methodologies.<br>► Articles outlining the challenges for merging social marketing to the promotion of health. | ► Articles reporting prevention issues and strategies.<br>► Articles that do not tackle health issues. |
| Context | ► Programmes and interventions designed for and implemented in any place and setting. | None |
| Others | ► English language.<br>► Original research articles and reviews published from 1 January 2010 onwards. | ► Books and books' chapters.<br>► Commentaries.<br>► Dissertations.<br>► Editorial notes.<br>► Essays.<br>► Proceedings.<br>► Scientific controversies.<br>► Working papers. |

will be extracted by two independent reviewers (MB and CS). Inter-rater reliability will be assessed using Cohen's Kappa statistic. Any disagreements that arise between the reviewers will be resolved through discussion and will be submitted to a third person if consensus cannot be established. Authors of papers will be contacted to request missing or additional data, where required.

In addition, gaps and limitations in the literature identified by the authors from the included articles will be noted carefully.

## Stage V: collating, summarising and reporting the results

Evidence from the data extracted will be synthesised and reported in seven non-mutually exclusive categories derived from the research questions and purposes of the scoping review: (i) development, (ii) effectiveness, (iii) programmes, (iv) interventions, (v) social marketing for health, (vi) promotion of cognitive health, (vii) challenges.

A narrative synthesis will provide a comprehensive overview of the main findings, gaps and future directions related to the scoping review objectives and questions.

## ETHICS AND DISSEMINATION

The scoping review will use a rigorous method to systematically search and map the literature on social marketing for health and on the promotion of cognitive health. Only articles published in peer-reviewed scientific journals will be included. Therefore, ethics approval will not be required for conducting this scoping review. At the international level, results from the full review will be disseminated in a peer-reviewed scientific journal to maximise their impact. At the national level, results will be disseminated with key stakeholders (healthcare providers and policy-makers) working in the fields of social marketing for health and of the promotion of (cognitive) health. This scoping review will be an important step towards research and practice on the promotion of cognitive health.

**Correction notice** This article has been corrected since it first published. Supplementary file has been updated.

**Contributors** MB: coordinated and conceptualised the project; ran a pilot test and developed the search strategy; developed the project methodology and initiated the first draft of the manuscript. CS: conceptualised the project; ran a pilot test and developed the search strategy. AK and CF: provided analytical and practical inputs for the implementation of active ageing policies in societies and healthcare settings. J-PS and CV: contributed to conceptualising the project. AK, CF, J-PS and CV: reviewed and commented on the manuscript.

**Competing interests** None declared.

**Patient consent for publication** Not applicable.

**Provenance and peer review** Not commissioned; externally peer reviewed.

terminology, drug names and drug dosages), and is not responsible for any error and/or omissions arising from translation and adaptation or otherwise.

**ORCID iD**
Mathilde Barbier http://orcid.org/0000-0002-4425-5705

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
