## [Reviewer comments · BMJ Open]

ARTICLE DETAILS

TITLE (PROVISIONAL)	Using social marketing for the promotion of cognitive health: a scoping review protocol.
AUTHORS	Barbier, Mathilde; Schulte, Caroline; Kornadt, Anna; Federspiel, Carine; Steinmetz, Jean-Paul; VÖGELE, Claus

VERSION 1 – REVIEW

REVIEWER	Tham, Rachel Australian Catholic University, Mary MacKillop Inst Health Res
REVIEW RETURNED	07-Apr-2021

GENERAL COMMENTS	This study protocol for a systematic scoping review is exploring a new field - the integration of social marketing with the promotion of cognitive health to enable better targeted interventions to be developed while using existing strategies that have been shown to be effective in other disciplines. The study protocol is clearly written and laid out in a logical sequence, backed up by existing and emerging literature in this field. The study protocol proposed in this manuscript appears to address all the key elements required for a scoping review. The only aspect that I think the authors may need to be clearer are the Search concepts and terms in Table 1. I understand that these terms are not exhaustive or finalised at this stage but for the protocol, accuracy of search terms should be reported. For example: 'market approach' should be 'market approach*'; 'tayloring' should be 'tailoring'; 'behaviour' and 'ageing' should be written to detect British and American English styles; 'skills' should be 'skill*'. 
---

REVIEWER	Howard Wilsher, Stephanie University of East Anglia, MED
REVIEW RETURNED	22-Apr-2021

GENERAL COMMENTS	Hello, I have some questions How do you define social marketing strategies? How are you defining cognitive health? This is a bit tricky of how cognitive health can be improved, and measured, through social marketing. There is no mention of statistics, and you say "qualitative", however, that gives the wrong impression unless you are looking at qualitative research. Perhaps it should be termed a narrative review? The search strategy is limited - short time frame. You are going to include Cochrane but thats a good place to start before searching to check for any previous work.
---

	The inclusion/exclusion criteria should be tighter. Are you including children or animals? Are you going to include all methodologies/publications? E.G. Commentaries.
--	--

VERSION 1 – AUTHOR RESPONSE

○ **Reviewer 1**

Reviewer 1:

I think the authors may need to be clearer are the Search concepts and terms in Table 1. I understand that these terms are not exhaustive or finalised at this stage but for the protocol, accuracy of search terms should be reported. For example: 'market approach' should be 'market approach*'; 'tailoring' should be 'tailoring'; 'behaviour' and 'ageing' should be written to detect British and American English styles; 'skills' should be 'skill*'.

Author:

On page 6, in Table 1, we have changed some terms and added new ones. Now you can find:

- “market* approach*” instead of “market* approach”;
- “marketing mix” instead of “market* mix**”
- “tailoring” instead of “market “tailoring”
- “social market**” instead of “social marketing”
- “health-promot*” instead of “health-promoting”
- “prevent**” instead of “prevention”
- “cognitive skill*” instead of “cognitive skills”.

We have also included the new terms: “marketing”, “health behavior”, “health”, and “healthy aging”.

○ **Reviewer 2**

1. Reviewer 2:

How do you define social marketing strategies?

Author:

On page 3 (fifth paragraph), we have included the following definition: “a social marketing strategy consists of a creation and administration of a marketing plan to effectively attract, motivate and retain an audience in performing a targeted prosocial behaviour.”

2. Reviewer 2:

How are you defining cognitive health?

Author:

On page 3 (first paragraph), we have defined cognitive health as “the ability to clearly think, learn and remember”⁷.

In addition, we have defined the promotion of cognitive health as “[enabling people to] increase control over their own [cognitive] health [...] [through] a wide range of social and environmental interventions”⁸.

3. Reviewer 2:

This is a bit tricky of how cognitive health can be improved, and measured, through social marketing.

Author:

On page 4 (first and second paragraph), several lines of explanation have been offered.

- The first paragraph provides a rationale on how cognitive health can be improved through social marketing, based on the Behaviour Change Wheel model (Michie, Atkins, & West, 2014). Thus, you can find: “social marketing strategies may concern implementation intentions to increase capability, and ultimately, motivation and participation of older people in behaviour promoting cognitive health. They may also concern the use of public media to increase opportunity for the older people who lack knowledge to get familiar with this domain of their health, and to identify available resources in their local area that will help them maintaining their cognitive health. A last but not least strategy may concern the use of a social identity approach, e.g. through positive social norms, to enhance the motivation of older people health-promoting behaviour with regard to their cognitive health.”
- With regard to how cognitive health can be measured through social marketing, the second paragraph states that: “efficacy will be measured directly by monitoring the number of people using the services (programs and interventions) designed to promote cognitive health. A cumulative positive effect can also be observed. If older people use these services more and feel empowered to do so, they will be potentially more motivated and committed to continue using these services, and hence to promote the services among their acquaintances and family.”

4. Reviewer 2:

There is no mention of statistics, and you say "qualitative", however, that gives the wrong impression unless you are looking at qualitative research. Perhaps it should be termed a narrative review?

Author:

On page 2, in the abstract, we have replaced the term “qualitative” with the term “narrative”.

5. Reviewer 2:

The search strategy is limited - short time frame. You are going to include Cochrane but that's a good place to start before searching to check for any previous work.

Author:

On page 5, in the methods section (paragraph "Stage II: Identifying relevant studies / Search strategy"), we have indicated that: "as a preliminary step, all the publications in English prior to 2010 will be checked electronically in the Cochrane Library."

6. Reviewer 2:

The inclusion/exclusion criteria should be tighter. Are you including children or animals? Are you going to include all methodologies/publications? E.G. Commentaries.

Author:

On page 6, in Table 2,

- as inclusion criteria, we have specified: "adults" instead of "persons", in the population section; "qualitative and quantitative methodologies", in the concept section; and "original research articles and reviews" instead of "articles", in the other section.
- as exclusion criteria, we have specified "people under the age of 18 years" and "animals", in the population section; and we have added "commentaries", "editorial notes", "essays", and "scientific controversies", in the other section.

We hope that the changes made will answer your questions and we hope the revised manuscript will better suit the Journal. We are open to consider further revisions. We thank you for your continued interest in our research.

VERSION 2 – REVIEW

REVIEWER	Tham, Rachel Australian Catholic University, Mary MacKillop Inst Health Res
REVIEW RETURNED	01-Sep-2021
GENERAL COMMENTS	The authors have addressed all the comments raised by the Editor and the Reviewers and have comprehensively described the background and methods for this scoping review which will lead to a systematic review. This is an important emerging area which has implications for a global ageing demographic utilising increased levels of social media and social marketing.